# scMD facilitates cell type deconvolution using single-cell DNA methylation references

Manqi Cai[1], Jingtian Zhou[2,3], Chris McKennan[4] & Jiebiao Wang [1,5✉]

The proliferation of single-cell RNA-sequencing data has led to the widespread use of cellular deconvolution, aiding the extraction of cell-type-specific information from extensive bulk data. However, those advances have been mostly limited to transcriptomic data. With recent developments in single-cell DNA methylation (scDNAm), there are emerging opportunities for deconvolving bulk DNAm data, particularly for solid tissues like brain that lack cell-type references. Due to technical limitations, current scDNAm sequences represent a small proportion of the whole genome for each single cell, and those detected regions differ across cells. This makes scDNAm data ultra-high dimensional and ultra-sparse. To deal with these challenges, we introduce scMD (single cell Methylation Deconvolution), a cellular deconvolution framework to reliably estimate cell type fractions from tissue-level DNAm data. To analyze large-scale complex scDNAm data, scMD employs a statistical approach to aggregate scDNAm data at the cell cluster level, identify cell-type marker DNAm sites, and create precise cell-type signature matrixes that surpass state-of-the-art sorted-cell or RNA-derived references. Through thorough benchmarking in several datasets, we demonstrate scMD's superior performance in estimating cellular fractions from bulk DNAm data. With scMD-estimated cellular fractions, we identify cell type fractions and cell type-specific differentially methylated cytosines associated with Alzheimer's disease.

[1] Department of Biostatistics, University of Pittsburgh, Pittsburgh, PA, USA. [2] Genomic Analysis Laboratory, The Salk Institute for Biological Studies, La Jolla, CA, USA. [3] Bioinformatics and Systems Biology Program, University of California, San Diego, CA, USA. [4] Department of Statistics, University of Pittsburgh, Pittsburgh, PA, USA. [5] Clinical and Translational Science Institute, University of Pittsburgh, Pittsburgh, PA, USA. ✉email: jbwang@pitt.edu

Tissue-level quantification of omics has gained popularity in the last decades because of its mature technology and affordable cost. Numerous studies on tissue-level omics, such as gene expression and DNA methylation (DNAm), provide rich resources to help answer interesting biological questions. However, bulk omics data are generated from a mixture of cells, meaning tissue-level analyses are often confounded by cellular heterogeneity, and cell type-specific (CTS) signals are obscured. While labor-intensive technologies such as flow cytometry and immunohistochemistry (IHC) can help measure cell type compositions, they are costly and more challenging for solid tissues[1]. As a cost-efficient alternative, in silico cellular deconvolution methods have been developed to recover the cell type composition of bulk omics data, allowing us to adjust for confounding cellular heterogeneity and infer CTS associations from bulk data[2–4].

Recent advances in single-cell technology have fueled numerous studies, leveraging high throughput single-cell RNA sequencing (scRNA-seq) as a reference to estimate cellular fractions in bulk RNA-seq data[5,6]. However, this progress in scRNA-seq stands in stark contrast to single-cell DNA methylation (scDNAm), which remains less studied. As a consequence, DNAm-based cell proportion estimates are often imprecise and can only be obtained for coarse cell types compared to RNA-based deconvolution. For example, deconvolving brain DNAm has been predominantly restricted to references derived from two cell types: neurons and non-neurons[7,8]. Two studies have worked on providing brain DNAm reference in better resolution other than two cell types. EpiSCORE[9] was proposed to deconvolve brain DNAm into six cell types. It employs scRNA-seq data to create a proxy signature for DNAm at the gene level. Specifically, EpiSCORE uses a scRNA-seq-derived reference to impute the DNAm at the promoter regions of marker genes and runs deconvolution based on these imputed signatures. However, not all CpGs in the promoter region are CTS, and EpiSCORE's imputation function mapping marker gene counts to promoter DNAm is not CTS. These compromise the cell type-specificity and accuracy of their DNAm signature, which are critical for the fidelity of deconvolution[10]. A more recent tool known as HiBED[11] integrates multiple sorted-cell DNAm references to deconvolve brain tissues into astrocytes, excitatory neurons, inhibitory neurons, microglia, oligodendrocytes, endothelial cells, and stromal cells. However, HiBED sources its references from different platforms such as 450K, EPIC, and whole-genome bisulfite sequencing (WGBS), leading to potential concerns regarding the handling of batch or platform effects when merging data from these sources. There is also a lack of sorted-cell references to extend it to more cell types.

Fortunately, scDNAm has been emerging in the last few years, especially for the brain[12–15]. The data exhibits strong cell type specificity, offering the potential to deconvolve tissue-level DNAm data. However, due to technical limitations, these methods usually detect only a small fraction of the genome in each single cell (~5% of all CpG sites), and the regions being detected could be highly variable between cells. Consequentially, the data is ultrahigh-dimensional and sparse, presenting considerable computational challenges.

To address these issues, we developed scMD (single cell Methylation Deconvolution), which uses scDNAm data to generate a high-quality DNAm reference and deconvolve bulk DNAm data. scMD leverages the strong cell type-specificity exhibited by scDNAm markers to perform high-resolution and accurate cellular deconvolution. Critically, scMD addresses the statistical and methodological hurdles that accompany scDNAm data, including its ultrahigh-dimensionality and sparsity, to identify cell-type marker CpGs and construct a signature that is amenable to bulk DNAm data. We use six real bulk DNAm datasets to illustrate scMD's superior performance over existing methods, where we show its ability to better estimate cellular fractions and infer Alzheimer's disease-related cell types. With scMD, we can complement bulk DNAm analyses with estimated cellular fractions to deconfound tissue-level analyses and enable CTS analyses.

## Results

**Overview of scMD**. Here we provide an overview of scMD, which uses scDNAm data to construct DNAm signatures amenable to bulk data and perform deconvolution (Fig. 1). The most challenging aspect of scDNAm is its high dimensionality and sparsity, which arises because only a small fraction (~5%) of the roughly 27 million DNAm sites are measured in each cell (Supplementary Table 1). The set of measured sites is cell-specific, meaning cell-type marker selection and signature matrix generation tools that require fully observed data, like those traditionally employed in scRNA-seq data[16,17], are not applicable in scDNAm. To address this, we subset sites observed in bulk data, e.g., CpGs on Illumina's 450k and EPIC arrays or in WGBS, and aggregate them across cells of the same type to obtain a much smaller and more computationally tractable cell cluster-level dataset. With methylated and unmethylated read counts, we then use Fisher's exact test to identify cell-type marker CpGs from cluster-level scDNAm data (Methods). This results in CTS p-values that compare one cell type with all other cell types. By conducting GREAT analysis[18] for our identified marker CpGs, we verified that most cell-type marker CpGs have the corresponding CTS functions (Supplementary Table 2 and Supplementary Data 1). After verifying the CTS functions of those identified cell type marker CpGs, we construct our signature matrix to be the beta values of marker sites in each cell type.

In contrast to existing DNAm-based deconvolution approaches that segregate brain tissue into coarse cell types (neurons and non-neurons)[7] or use RNA-derived[19] or sorted-cell[11] signatures, our method takes advantage of recent advancements in brain scDNAm resources[13,15] to construct the first brain scDNAm signature matrices encompassing seven distinct cell types: astrocytes, endothelial cells, excitatory neurons, inhibitory neurons, microglia, oligodendrocytes, and oligodendrocyte progenitor cells (OPC) (Fig. 1). After constructing the DNAm signature matched with the target bulk DNAm data, we employ our previously developed robust and precise cellular deconvolution method, EnsDeconv[10] (ensemble deconvolution). For this work, we integrate different references, data transformations, and deconvolution algorithms. EnsDeconv incorporates a CTS robust regression to examine the viable combinations of the factors related to cellular deconvolution, ensuring optimal cellular fraction estimation.

**Validating scMD using sorted-cell data**. We assessed the accuracy of scMD in deconvolution using three different sorted-cell datasets derived from various DNAm platforms. This evaluation allowed us to understand scMD's performance across multiple technologies and gauge its proficiency in accurately deconvolving different purified-cell samples. We first tested scMD with the dataset from Mendizabal et al.[20], which quantified WGBS DNAm from sorted neurons (NeuN+) samples and OLIG2+ samples that indicate oligodendrocytes and OPC. We then utilized the datasets from Guintivano et al.[8] and Gasparoni et al.[21], both containing sorted-cell DNAm samples from NeuN+ and non-neurons (NeuN−). All samples from the three datasets have definitive fractions of non-neurons, neurons, or the sum of oligodendrocytes and OPC. These datasets provided an opportunity

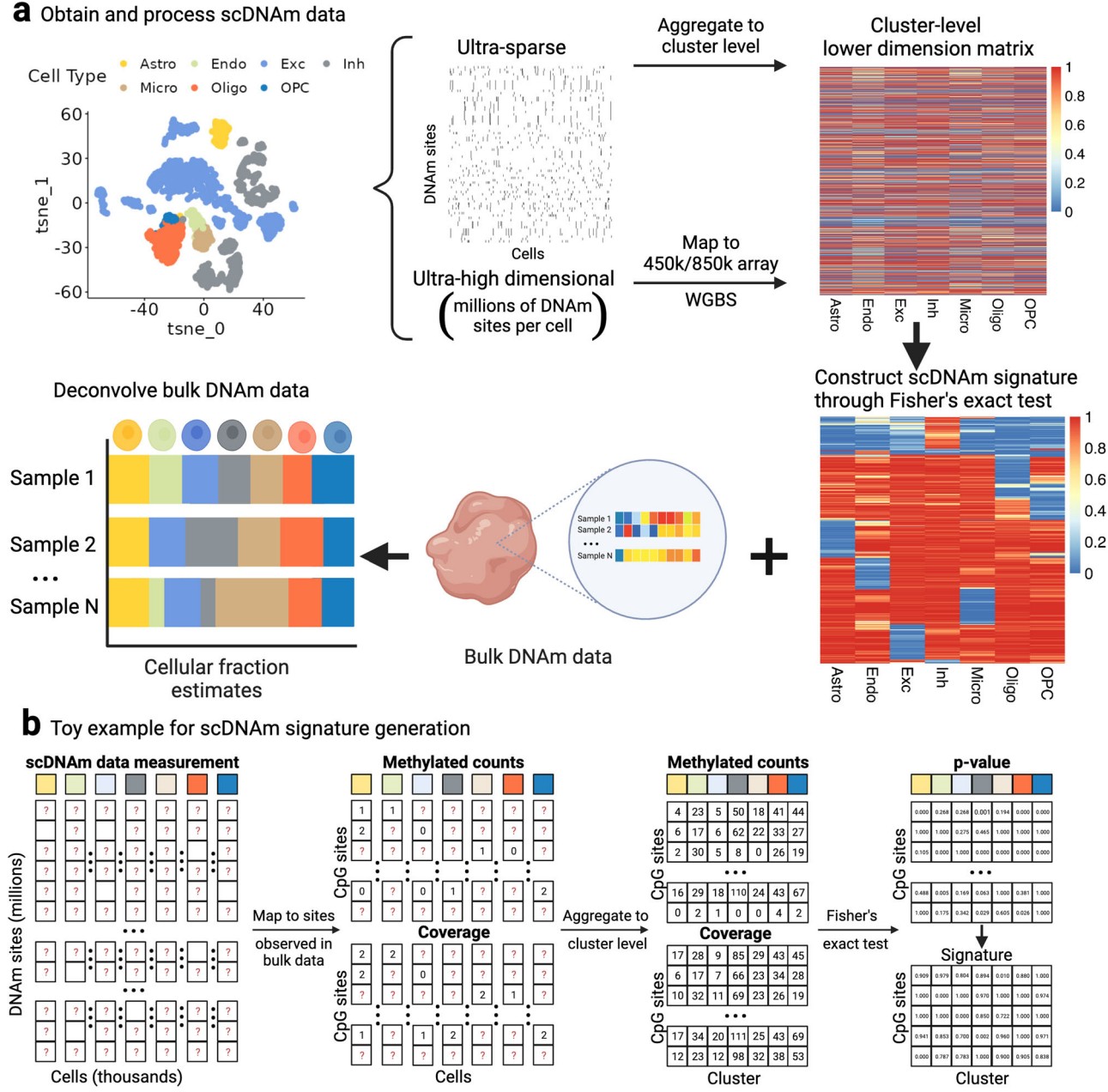

**Fig. 1 scMD workflow. a** Schematic representation of the proposed scMD framework. First, with single-cell DNA methylation (scDNAm) data and cell cluster labels, we filter the scDNAm data to match bulk DNAm sites (450k, EPIC array, or WGBS), reducing the dimensionality. Second, we aggregate the scDNAm data at the cluster level to mitigate the issue of prevalent missingness. Third, Fisher's exact test is utilized to identify marker CpGs by comparing each cell type against all other cell types. Finally, based on the resulting *p*-values, a distinctive scDNAm signature is constructed. Here we show seven cell types: astrocytes (Astro), endothelial cells (Endo), excitatory neurons (ExN), inhibitory neurons (InN), microglia (Micro), oligodendrocytes (Oligo), and oligodendrocyte precursor cells (OPC). **b** Detailed demonstration of building DNAm signature matrix from high-dimensional and sparse scDNAm data. Question marks denote missing data. Column annotations represent cell types. This figure is created with BioRender.com.

to accurately measure scMD's performance in identifying and distinguishing between various major brain cell types. Further details about the validation datasets and the approaches employed for evaluating the performance of scMD are outlined in the Methods section and Supplementary Table 3.

We carried out a comparative analysis of scMD with EpiSCORE[19] and HiBED[11]. Tested on the Mendizabal dataset[20], scMD almost perfectly fits the data, accurately deconvolving the neuron and oligodendrocyte samples (Fig. 2a). Compared to EpiSCORE, which has difficulty differentiating OLIG2+ and other cell types, our proposed method excelled in effectively

identifying sorted-cell samples as their corresponding cell types. This suggests that scMD can effectively harness signals from the originally sparse scDNAm data. For a fair comparison, we did not compare HiBED here, since HiBED utilized the Mendizabal dataset to construct its reference panel. We also evaluated scMD's accuracy in deconvolving 450k array-based samples available from Guintivano et al.[8] (Fig. 2b and Supplementary Fig. 1) and Gasparoni et al.[21] (Fig. 2c), which underscore scMD's ability to accurately deconvolve both NeuN+ and NeuN- samples, thereby demonstrating its versatility and efficiency in brain cell deconvolution.

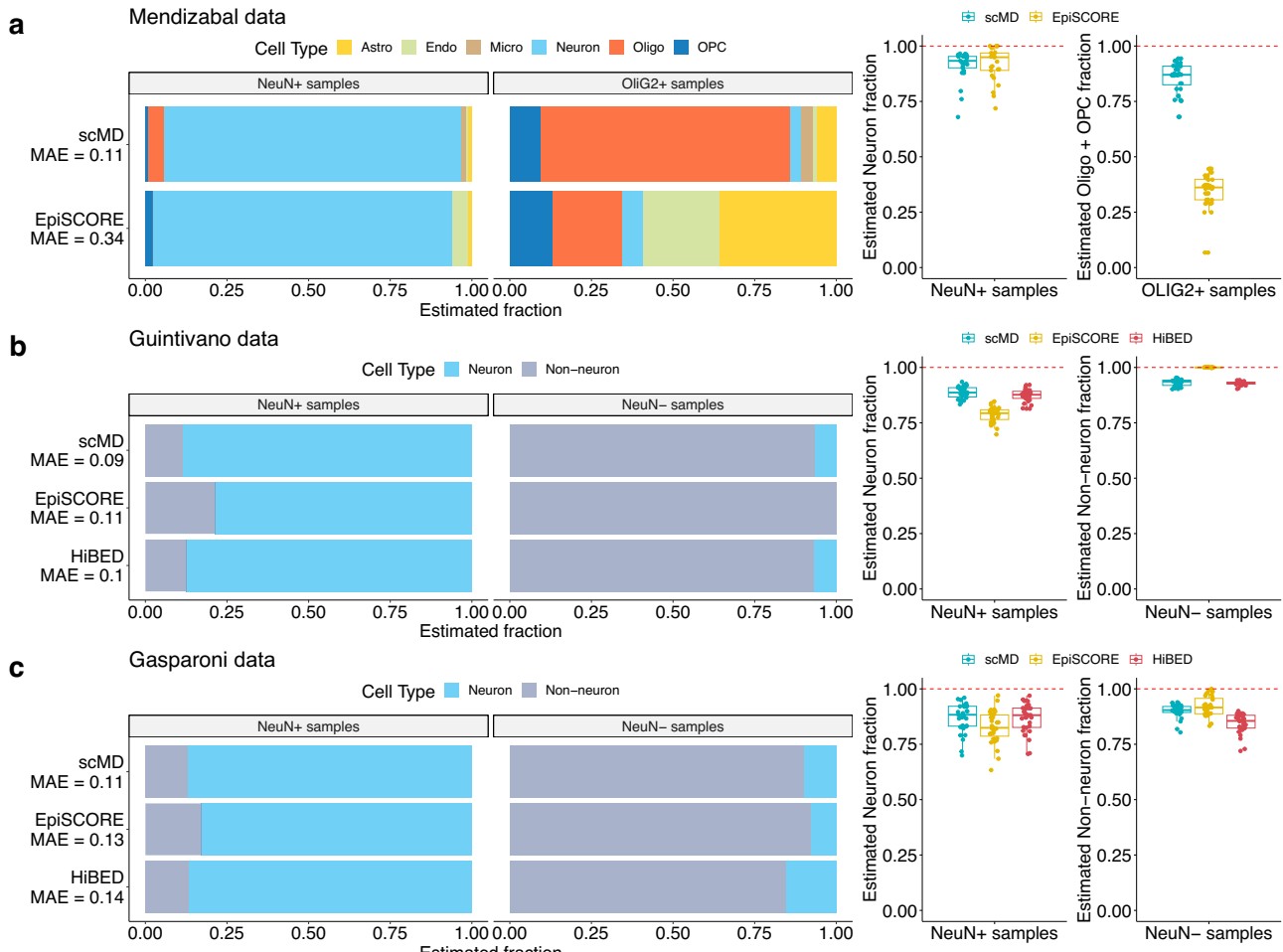

**Fig. 2 Validating scMD with sorted-cell data. a** Validation on Mendizabal et al.[20]. Bar plots of mean estimated cellular fractions across NeuN+ and OLIG2+ samples using scMD and EpiSCORE. Different cell types are annotated with different colors. Box plots of cellular fractions in sorted NeuN+ ($n = 25$ biologically independent samples) and OLIG2+ samples ($n = 20$ biologically independent samples) are shown on the right. Different colors represent different methods. **b** Validation on Guintivano et al.[8]'s NeuN+ ($n = 29$ biologically independent samples) and NeuN- samples ($n = 29$ biologically independent samples). A comparison of scMD, EpiSCORE, and HiBED is presented. **c** Validation on Gasparoni et al.[21]'s NeuN+ ($n = 31$ biologically independent samples) and NeuN- samples ($n = 31$ biologically independent samples). This section also contrasts the results obtained from scMD, EpiSCORE, and HiBED. Note that for benchmarking, we aggregated the fraction estimates of cell subtypes to generate the fractions of broader cell types. For a fair comparison, HiBED is not compared using Mendizabal et al.[20], which was utilized by HiBED as a reference. For all box plots, the median is indicated by the central line, quartiles by the box edges, and whiskers extend to 1.5 times the interquartile range, with outliers plotted individually. The reference line where the estimated fraction equals one is plotted as a red dashed line. Source data can be found in Supplementary Data 5–7.

**scMD accurately estimates cellular fractions in cerebral cortex.** To gain deeper insights into the performance of scMD, we conducted a comprehensive comparison of scMD with various other deconvolution methods using real bulk data with IHC-measured cell counts of four cell types from cerebral cortex samples that were part of the Religious Orders Study (ROS)[22]. We also used our signatures as input into existing deconvolution methods to demonstrate the importance of our signature matrices and illustrate the fidelity of EnsDeconv when applied to DNAm.

On average, scMD significantly outperforms EpiSCORE and HiBED (Fig. 3a). Especially EpiSCORE exhibits a low correlation with the measured fractions of microglia and astrocytes. This is because EpiSCORE consistently estimates microglia fractions to be zero and tends to overestimate astrocyte fractions (Fig. 3c). On the other hand, HiBED appears to underestimate astrocytes and overestimate oligodendrocytes (Fig. 3d). Its estimated astrocyte fractions have a strong negative correlation with IHC-measured fractions. In contrast, the fractions estimated by scMD and those measured through IHC are consistent, especially for astrocytes

and microglia (Fig. 3a, b). This alignment underscores the importance of accurately estimating microglia fractions, as microglia is a crucial brain cell type implicated in multiple diseases, such as Alzheimer's disease[23]. We further calculated mean absolute error (MAE) for the main datasets we used and conducted the Friedman-Nemenyi posthoc test and one-sided Diebold-Mariano tests to demonstrate that scMD outperforms existing methods (Supplementary Tables 4–6). Results also show that provided they utilize our scDNAm-based signature, existing deconvolution methods also outperform EpiSCORE (Fig. 3a), thereby further illustrating the accuracy of our signatures. We do note, however, that scMD, which utilizes EnsDeconv to perform deconvolution, outperforms all methods.

**Consistent cellular fractions estimated from DNAm and mRNA.** While it is ideal to validate scMD with measured cell counts, the resources are limited to major cell types and small sample sizes given the challenges of counting cell types in solid

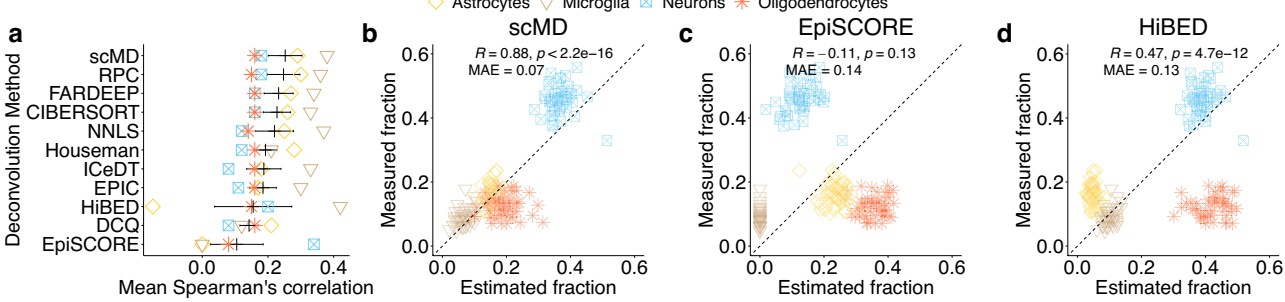

**Fig. 3 Cell type deconvolution results for ROS data. a** Benchmarking of scMD and other deconvolution methods on ROS data. EpiSCORE uses its RNA-derived reference. HiBED employs its sorted-cell DNAm reference. All other methods adopt our scDNAm references. For scMD, EpiSCORE, and HiBED, each dot denotes one correlation for each cell type. For other methods, each dot represents the average of Spearman's correlation across scenarios in each cell type. A scenario is defined as a particular setting with a specific deconvolution method and reference dataset. The black vertical line shows the mean of the average Spearman's correlation across scenarios, and the error bars present means ± standard error of the mean. **b–d** Scatterplots illustrate the relationship between the estimated fractions of scMD, EpiSCORE and HiBED (x-axis) against the corresponding fractions measured using immunohistochemistry (IHC) in ROS data (y-axis). This comparison involves 49 biologically independent samples. The correlation is calculated by pooling fractions across cell types. Note that for benchmarking, we aggregated scMD's and HiBED's fraction estimates of neuronal subtypes to generate the neuronal fractions. Source data can be found in Supplementary Data 8.

tissues like the brain. Instead, the deconvolution of RNA-seq data has been well benchmarked and thus can be used as "ground truth"[10]. In addition to the ROS data, we further validated scMD in more cell types and a different platform with the dataset from Markunas et al.[24], which quantified paired DNAm of Illumina EPIC arrays and RNA-seq bulk data from the nucleus accumbens (NAc) of 211 individuals. Even though we do not have measured cell counts, the cellular fractions are available from deconvolving paired mRNA data. The intuition is that if we possess paired DNAm and RNA bulk data from the same tissue samples, we should observe high concordance between the estimated cellular fractions from these two omics types, given that there is a single true cellular composition for a tissue sample.

With the above rationale, we first estimated cellular fractions using RNA data[10] as the "ground truth" of cellular fractions for benchmarking. Equipped with our newly constructed signature matrices, we deconvolved the NAc bulk DNAm and RNA data and examined the deconvolution results between these two omics types. We obtained a strong correlation between estimated RNA- and DNAm-based fractions when we employed scMD to deconvolve DNAm samples and EnsDeconv on RNA samples (Fig. 4a). Except for OPC, all correlations exhibited were close to or above 0.5. The correlation was especially noticeable among major cell types such as neurons and oligodendrocytes, where correlations of 0.82 and 0.89 were observed. Furthermore, the correlation remained high (0.73) even for the less common endothelial cells. In contrast, when using EpiSCORE to infer cellular fractions from DNAm, the correlations are lower than those of scMD across all cell types (Fig. 4b). Notably, EpiSCORE consistently estimates microglia fractions to be ~0, and its correlations for astrocytes and OPC are both negative. When we employed HiBED for inference, we observed a high correlation for neurons and oligodendrocytes. However, for other cell types, the correlations are substantially lower than those of scMD. Additionally, HiBED tends to overestimate oligodendrocytes and microglia and underestimate astrocytes and endothelial, showing higher MAE than that of scMD (Fig. 4c).

**scMD identifies cell types associated with Alzheimer's disease.** To demonstrate the utility of scMD-estimated cellular fractions, we tested their associations with clinical phenotypes related to Alzheimer's disease (AD). We utilized the brain DNAm data from Mount

Sinai Brain Bank (MSBB), which also collected variables such as age, Clinical Dementia Rating (CDR), the Consortium to Establish a Registry for Alzheimer's Disease (CERAD) score, and Braak stage. The CDR was employed as an assessment tool to evaluate dementia and cognitive status, assigning ratings on a scale of 0 to 5, which correspond to escalating levels of severity in pathology[25]. The CERAD score is a four-level semi-quantitative measure of neuritic plaques. Braak stage is a widely used classification system indicating the progression of AD and categorizes the advancement of neurofibrillary tangles and amyloid plaques in the brain, with stages ranging from 0 to 6, representing increasing levels of pathology severity[26,27].

Given the neurodegeneration that accompanies AD, comparing cell-type fractions across age and various AD phenotypes is therefore of scientific interest. We conducted a comprehensive study examining the correlation of various phenotypes in MSBB with estimated cellular fractions using scMD, EpiSCORE, and HiBED (Fig. 5a and Supplementary Table 7). Consistent with previous studies[28,29], scMD detected a significant decrease in OPC with aging. While HiBED identified a significant decrease in inhibitory neurons, EpiSCORE failed to associate any cell types with age. Among the three AD-related phenotypes, we found the most differential fraction signals in clinical dementia rating. With scMD, we observed significantly increased fractions of microglia and oligodendrocytes and decreased fractions of OPC and excitatory neurons.

In contrast, EpiSCORE highlighted only a significant rise in astrocytes and oligodendrocytes and a decline in neurons. Meanwhile, HiBED identified increased microglia and oligodendrocytes and decreased excitatory neurons.

Interestingly, as two aspects of AD, neuritic plaques and neurofibrillary tangles show strikingly different differential fraction results. scMD, EpiSCORE, and HiBED all did not identify any cell types associated with neuritic plaques (as indicated by the CERAD score), but there are some cell types associated with neurofibrillary tangles (as measured by Braak score). For instance, scMD and HiBED-estimated microglia and excitatory neuron proportions increase and decrease as the Braak stage increases, respectively (Fig. 5b–h and Supplementary Fig. 2), and inhibitory neuron proportions exhibit little change. The observed increase in microglia proportions suggests an enhanced immune response and neuroinflammation, which are known to be critical in neurodegenerative disorders like AD[23]. Additionally, the substantial decline in excitatory neurons is compelling.

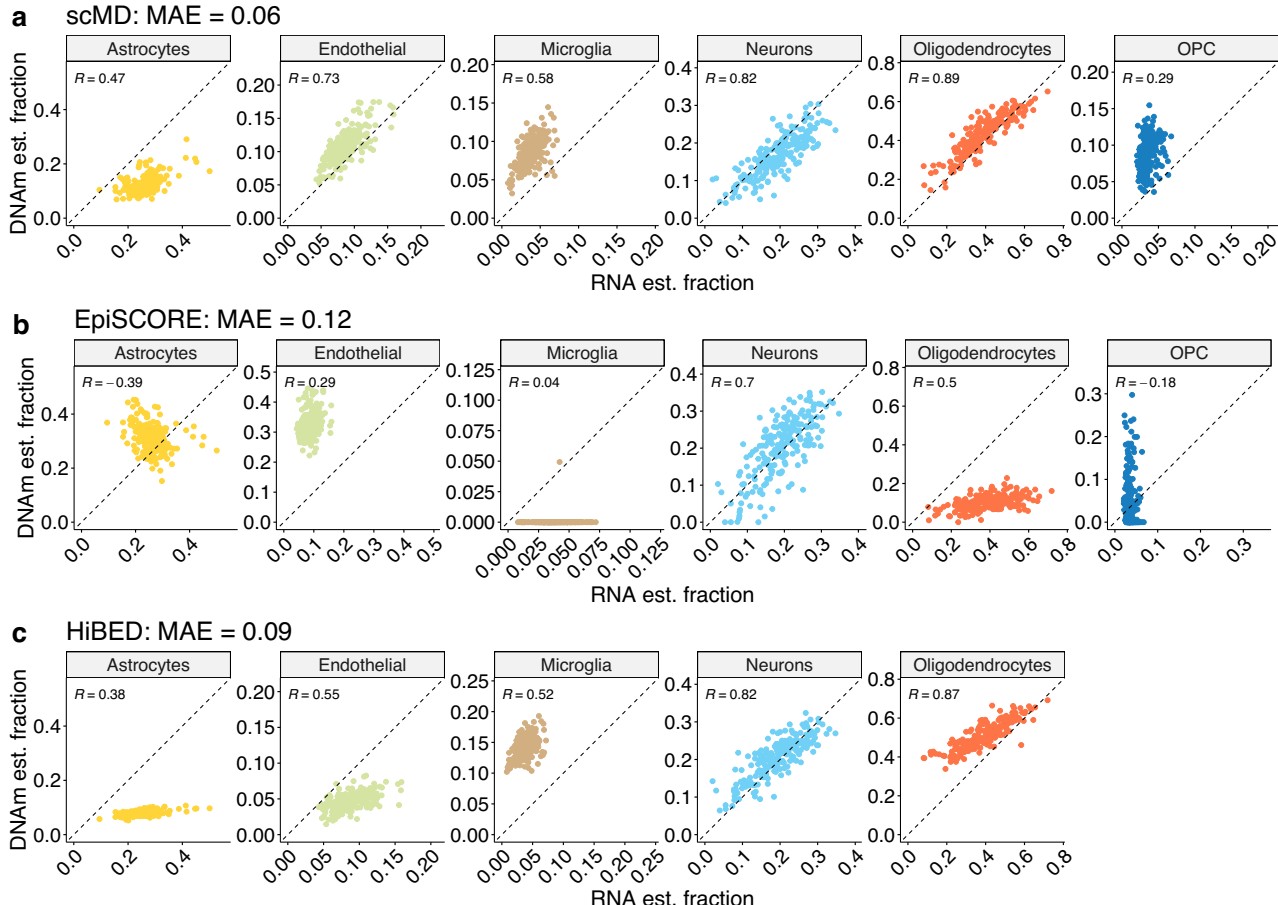

**Fig. 4 Cell type deconvolution results for NAc data.** Deconvolution of bulk NAc data ($n = 211$ biologically independent samples): comparison of cell type estimation using scMD (**a**), EpiSCORE (**b**), and HiBED (**c**). Scatter plots showcase the relationship between the estimated cell fractions from RNA data (x-axis) using EnsDeconv and DNAm data (y-axis). scMD and EpiSCORE estimate six cell types since NAc does not have excitatory neurons, while HiBED does not estimate OPC.

Excitatory neurons play a crucial role in signal transmission and neural communication within the brain. The reduction in their cell count implies potential disruptions in synaptic activity and impaired neuronal function in affected brain regions. These findings align with previous research emphasizing neuronal loss as a sign of neurodegenerative disease.

Similarly, we also used EpiSCORE to estimate cell fractions from MSBB data and identified cell types associated with AD (Supplementary Fig. 3). EpiSCORE identified oligodendrocytes and OPC associated with the Braak score. Consistent with Fig. 4b, EpiSCORE estimates microglia proportions to be almost all zero and therefore not able to infer a significant correlation between their proportions and the Braak stage. While the decrease in neurons among AD patients is confirmed with EpiSCORE (Supplementary Fig. 3), it lacks the resolution to show the decrease is primarily driven by excitatory neurons since it does not estimate neuronal subtypes. Furthermore, scMD-estimated cellular fractions enable CTS differential methylation analyses. We used CellDMC[30] to identify cell type-specific differentially methylated cytosines (CTS-DMCs) (Supplementary Table 8 and Supplementary Data 2). With scMD-estimated cellular fractions, we identified 22 CTS-DMCs in microglia associated with age (Fig. 5i) and 57 DMCs in OPC with FDR < 0.05. Notably, among the most significant CpGs in microglia, cg18574144 is within the gene body of *THOP1*, which is currently under investigation as a potential biomarker for Alzheimer's disease[31]. For CDR, we detected 195 DMCs in astrocytes. We also identified dozens of DMCs in inhibitory

neurons for Braak score (neurofibrillary tangles). We also found some CTS-DMCs coincide with AD GWAS loci or nearby regions (Supplementary Data 3 and Supplementary Table 9). For instance, cg23393368, a DMC in astrocytes associated with CDR dementia staging, is close to rs4817090, an AD GWAS SNP mapped to gene *APP* that is important for AD.

There is a relevant study[32] that also presented DMCs in association with Braak scores using the MSBB dataset. They identified a total of 236 significant DMCs at the tissue level. We conducted a correlation test between the test statistics from our CTS-DMC analysis and those reported in their study, and most cell types show concordance (Supplementary Fig. 4). Notably, the bulk data analysis adjusted for neuron fraction. However, their adjustment did not consider the non-neuron fractions, nor the finer granularity of cell types that we employed in our analysis. This limitation may lead to the observed suboptimal correlations in some cell types that may be confounded in bulk data analysis by cellular fractions. We also conducted an enrichment analysis using gprofiler2 and have included the results in Supplementary Data 4. We identified some interesting enriched pathways, such as the AD pathway for endothelial DMCs associated with CDR.

## Discussion
The scMD method we developed presents a considerable advancement in the ability to analyze and understand the cellular heterogeneity of the brain at the molecular level using DNAm

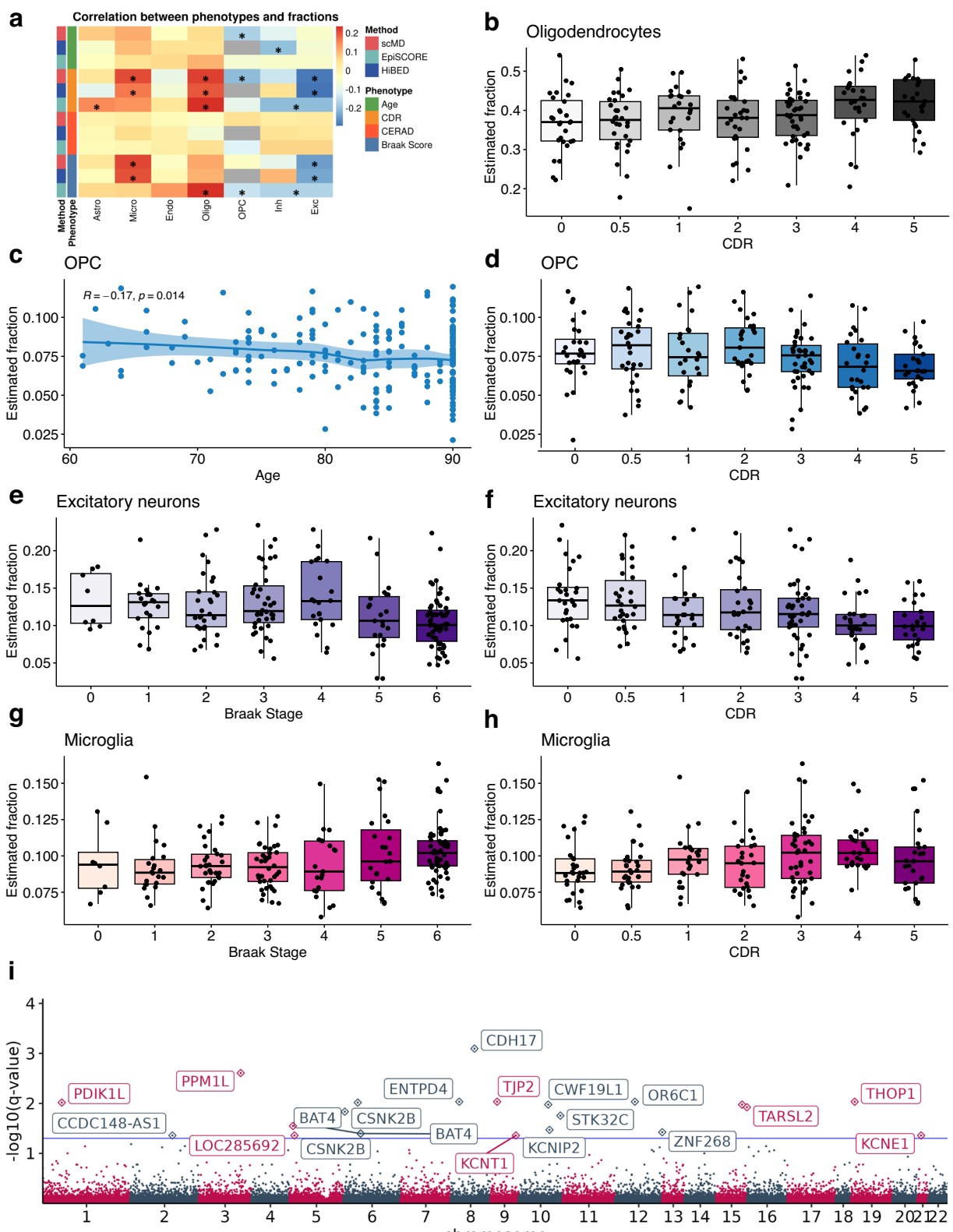

**Fig. 5 Cell type deconvolution results and CTS-DMC analysis for MSBB data.** Identified differential cellular fractions and methylated cytosines with the Mount Sinai Brain Bank (MSBB) data ($n = 201$ biologically independent samples). **a** Correlation between cellular fractions and age and AD phenotypes. * $p$-value < 0.05. **b–h** scMD identified pairs of phenotypes and cellular fractions. In Figure **c**, the scatterplot is presented with a shaded area representing the confidence interval around the LOESS smooth line. The results of HiBED and EpiSCORE are shown in Supplementary Figs. 2 and 3. **i** Differentially methylated cytosines in microglia associated with aging using scMD estimated cellular fractions. Significant CpGs are annotated to genes if available. For all box plots, the median is indicated by the central line, quartiles by the box edges, and whiskers extend to 1.5 times the interquartile range, with outliers plotted individually. Source data for Figure **b–h** can be found in Supplementary Data 9.

data. By constructing signature matrices for seven distinct brain cell types, scMD offers a much finer level of detail than previous deconvolution methods and has the potential to extend to more cell types with detailed scDNAm references. scMD goes beyond existing approaches by effectively leveraging recent advancements in scDNAm data resources, bridging the gap between single-cell and bulk DNAm data. This utilization of single-cell data in the generation of our signature matrices captures the intrinsic cellular heterogeneity of the brain, which is an important consideration in studying various brain-related diseases and conditions.

The accuracy of scMD is reflected in its high performance in various validation studies across different DNAm platforms. First, scMD consistently outperformed other approaches in the deconvolution of purified-cell and bulk datasets, highlighting its robustness and potential for widespread applications. By comparing scMD with ten existing deconvolution methods, we showed that it has higher concordance and lower mean absolute error when comparing estimated and ground truth fractions. Furthermore, scMD demonstrated high concordance between the RNA-estimated fractions and DNAm-estimated fractions, suggesting that scMD is successful in capturing useful signals from the original sparse scDNAm data. Lastly, we showed that scMD can precisely identify the fractions of microglia and excitatory neurons associated with AD. Via CTS-DMC analyses with various AD-related phenotypes, we demonstrated the usage of scMD in downstream analyses and validated that some CTS DMCs coincide with known AD GWAS loci.

Despite the evident promise shown by scMD, it is essential to recognize certain challenges and limitations that may warrant future work. First, there may be a computational burden presented by processing scDNAm data due to its high dimensionality. The massive volumes of raw data require computational resources to process and reduce storage and memory. However, given the availability of scDNAm data on public online platforms, we have mitigated this issue through our parallel computation approach. This method allows for the rapid and efficient processing of many cells at once during the construction of a cell-type signature matrix. Another potential issue lies in the performance of the model being contingent on the quality and scope of the reference single-cell data used to build the signature matrices. Despite incorporating diverse sources of data to create our matrices, representation of certain cell types, particularly rare ones, may be limited. Furthermore, the method's efficacy in tissues and conditions not represented in the training data awaits further evaluation. As scDNAm is increasingly applied to various tissue types, we anticipate a broadened use of scMD beyond the brain.

In summary, we present a robust and versatile tool for researchers to deconvolve bulk DNAm data with scDNAm references. By offering more accurate, detailed, and efficient analyses of brain cell composition from DNAm data, our method facilitates deeper exploration into the molecular aspects of brain function and pathology. In our future work, we plan to refine and expand the capabilities of scMD. We aim to incorporate additional cell types and explore various tissue types with the expansion of scDNAm. Additionally, we aim to integrate other omics data to deepen our understanding of cellular heterogeneity in the brain and other tissues.

## Methods

### Details of the proposed scMD framework

*Processing scDNAm data.* Our goal is to build high-quality DNAm references using scDNAm technology. The initial step towards achieving this objective involves harmonizing the differences in DNAm technologies between traditional bulk DNAm data and the noisier scDNAm data. Compared to bulk DNAm data, scDNAm is significantly sparser (with 95% missing data) and higher in dimensionality. The challenge is to bridge the gap between the dimensionality of the traditional bulk DNAm data and that of the scDNAm data. Traditional bulk DNAm data typically use arrays of 450k or EPIC CpG sites, while scDNAm is characterized by its sparse quantification across billions of genomic locations (CG + CH). To address this issue, we employ a strategy that involves subsetting the DNAm sites measured with 450k/EPIC arrays or WGBS. This approach serves twofold. Firstly, it accelerates the overall process. Secondly, it simplifies the process of identifying marker CpGs, which are crucial for various analyses in the field of epigenetics. By employing this technique, we can successfully reduce the dimensionality of scDNAm data to make it comparable to its bulk counterpart. After achieving a reduced-dimension scDNAm dataset, the subsequent step is to derive CTS $p$-values to identify marker CpGs. Given the inherent sparsity of scDNAm data, characterized by prevalent (95%) missing values, it is not feasible to identify specific markers and construct signatures in the same manner as with scRNA-seq data.

To generate a DNAm signature matrix akin to a reference, we first aggregated the methylated counts and coverage of DNAm of each cell type:

$$mc_{ik} = \sum_{j=1}^{N_k} mc_{ijk},\ cov_{ik} = \sum_{j=1}^{N_k} cov_{ijk},$$

where $mc_{ik}$ represents the cumulative count of all methylcytosine for the $i$th CpG site and $k$th cell type, $N_k$ denotes the number of cells belonging to the cell type $k$, and $cov_{ik}$ is the total cytosine basecalls, incorporating both methylcytosine and unmethylcytosine for the $i$th DNAm site and $k$th cell type. Subsequently, we carry out two-sided Fisher's exact tests for each cell type across all DNAm sites. To differentiate cell type $k$ from all other cell types for a given DNAm site $i$, we formulated Table 1.

The $p$-value of Fisher's exact test corresponding to cell type $k$ at the DNAm site $i$ is derived as

$$p_{ik} = \frac{\left(\sum_k M_{ik}\right)!\left(\sum_k U_{ik}\right)!\left(\sum_{k'\neq k} cov_{ik}\right)!\left(cov_{ik}\right)!}{(M_{ik})!\left(\sum_{k'\neq k} M_{ik'}\right)!(U_{ik})!\left(\sum_{k'\neq k} U_{ik'}\right)!\left(\sum_k cov_{ik}\right)!}.$$

Once we calculated the CTS $p$-values for each cell type across all DNAm sites, we arranged these values in ascending order. Based on a detailed evaluation of two sorted-cell datasets, we selected the top 100 marker DNAm sites for each cell type based on their $p$-values (Supplementary Fig. 5). This aligns with existing methods, such as minfi[7], which opted to select the top 100 differentially methylated marker DNAm sites per cell type. This strategic selection offers dual benefits. It not only accelerates the deconvolution process, making the computational burden manageable for extensive bulk DNAm datasets, especially for WGBS bulk DNAm data, but also

---

**Table 1 The 2 × 2 contingency table of Fisher's exact test for comparing cell type $k$ with other cell types for $i$th CpG.**

|  | Cell type $k$ | Other cell types | Row total |
|---|---|---|---|
| Methylcytosine | $M_{ik} = mc_{ik}$ | $\sum_{k'\neq k} M_{ik'} = \sum_{k'\neq k} mc_{ik'}$ | $\sum_k M_{ik}$ |
| Unmethylcytosine | $U_{ik} = cov_{ik} - mc_{ik}$ | $\sum_{k'\neq k} U_{ik'} = \sum_{k'\neq k} cov_{ik'} - \sum_{k'\neq k} mc_{ik'}$ | $\sum_k U_{ik}$ |
| Column total | $cov_{ik}$ | $\sum_{k'\neq k} cov_{ik'}$ | $\sum_k cov_{ik}$ |

enhances accessibility for the broader scientific community. By reducing computational costs, our approach alleviates the challenges for researchers, particularly those in resource-constrained settings, when handling large datasets.

Here we use an example to illustrate how scMD handles the large-scale raw scDNAm data. The original compressed raw scDNAm files for more than 4200 nuclei from Lee et al.[13] totaled a substantial 717 GB. After filtering for only CpG sites, the data was condensed to 183.2 GB. Further refinement at the cluster level reduced the data size to a more manageable 17.4 GB before loading into the R environment. Through the application of parallel computation across 20 nodes, we were able to generate an EPIC-based signature within ~10 min. Supplementary Table 1 provides detailed information on the number of DNAm sites before and after processing. To offer detailed instructions for scMD, we outline the algorithm in Supplementary Algorithm 1. It includes five steps from reducing dimensions, identifying marker CpGs, constructing scDNAm signature, to estimating cellular fractions with ensemble deconvolution (EnsDeconv).

*Ensemble deconvolution.* After obtaining the signature matrix from scDNAm data, the subsequent crucial step involves performing deconvolution on DNAm datasets. To accomplish this, we employed our previously developed method EnsDeconv[10]. In essence, EnsDeconv represents a deconvolution technique that draws inspiration from ensemble learning, wherein the outputs of multiple deconvolution algorithms are combined to achieve enhanced estimation accuracy. EnsDeconv focuses on important factors such as the choice of reference datasets, data transformations, and deconvolution methods. EnsDeconv implements CTS robust regression to synthesize results from different deconvolution settings, resulting in more robust and accurate results than randomly choosing one setting. Taking into account all possible combinations of the aforementioned factors in deconvolution, we leveraged ensemble learning to generate $\hat{P}_1, \ldots, \hat{P}_D$, representing the estimated cellular proportions from each of the $D$ scenarios. In this context, we define a scenario as a specific setting with a particular reference dataset, transformations approach, and deconvolution method. We treat the ensemble learning problem as a robust regression problem:

$$\underset{\boldsymbol{W}_1, \ldots, \boldsymbol{W}_K \in [0,1]^S}{\operatorname{argmin}} \sum_d \sum_{k=1}^K \| \hat{\boldsymbol{W}}_{dk} - \boldsymbol{W}_k \|_2,$$
$$(\boldsymbol{W}_1, \cdots, \boldsymbol{W}_K)\boldsymbol{1}_K = \boldsymbol{1}_S$$

where $\boldsymbol{W}_k$ denotes the $k$-th cell type's ensemble fraction for $S$ samples, $\hat{\boldsymbol{W}}_{dk}$ represents the estimate for the $k$-th cell type fraction in the $d$-th deconvolution scenario, and $\| \boldsymbol{v} \|_2 = (\sum_i \boldsymbol{v}_i^2)^{1/2}$ is the vector equivalent of absolute deviation.

In this study, we utilized two scDNAm references, Lee et al.[13] and Tian et al.[15], to implement the EnsDeconv approach. In terms of data transformations, scDNAm adopts both beta-value and M-value transformations. In addition, our implementation of EnsDeconv incorporated nine diverse deconvolution methods, each founded on unique theoretical bases and specifically designed for various purposes. A portion of these techniques was originally developed for deconvolving bulk DNAm data, e.g., quadratic programming[33] and robust partial correlations (RPC)[34]. In parallel, we also integrated several deconvolution methods primarily designed for RNA-seq experiments. These included the robust regression technique from FARDEEP[35], support-vector regression from CIBERSORT[36], the penalized regression method with elastic net regularization featured in DCQ[37], a log-normal model from ICeDT[38], and non-negative least squares (NNLS).

## DNA methylation datasets

*Brain scDNAm datasets.* In this study, we began by generating a reference for scDNAm using snmC-seq data obtained from Lee et al.[13]. The dataset utilized in this study is comprised of 4,238 single human brain prefrontal cortex cells, enabling the simultaneous capture of chromatin organization and DNA methylation information. The scDNAm data was downloaded from the GEO database (GSE130711). To ensure data quality, we utilized the cell-type annotation provided by the authors and excluded any cells marked as outliers, resulting in a total of 4,234 cells for further analysis. The distribution of cell-type annotations in the remaining dataset consisted of 670 inhibitory neurons (InN), 945 excitatory neurons (ExN), 1250 oligodendrocytes (Oligo), 449 astrocytes (Astro), 416 microglial cells (Micro), 315 endothelial cells (Endo), and 189 oligodendrocyte progenitor cells (OPC). To map the scDNAm data to the DNAm sites in the bulk DNAm dataset, we specifically considered cytosines in the CG context while excluding those in the CH context. Additionally, we incorporated data from a newly collected dataset Tian et al.[15], which employed snmC-seq3 technology to profile whole-genome DNAm data. We obtained the cluster-level data from the frontal cortex for the same seven cell types as Lee et al.[13].

*Sorted-cell brain DNAm datasets for validation.* Descriptions of DNAm validation datasets used in this part are summarized in Supplementary Table 3. In order to assess the accuracy of our signature matrices, we used three sorted-cell datasets that contained either sorted neuron samples and non-neuron samples or oligodendrocyte samples as validation datasets. The first dataset Mendizabal[20] is a whole-genome bisulfite sequencing (WGBS) postmortem human brain dataset. It is composed of two cell populations: NeuN+ and OLIG2+. We focus on healthy controls, and the sample size is 25 and 20 respectively for the two cell types. The data is downloaded from GEO (GSE108066). The second dataset Guintivano[8] is an Illumina Human 450k Methylation dataset and profiled in the postmortem frontal cortex of two different cellular populations (NeuN+ vs. NeuN-) generated from 29 individuals using flow sorting. Additionally, Guintivano contains 9 mixed samples, which are empirical blends of neuronal and glial DNA in increments of 10%. We downloaded the Guintivano data through the Bioconductor package FlowSorted.DLPFC.450k. The third dataset Gasparoni[21] is an Illumina Human 450k Methylation dataset that contains 62 sorted-cell frontal cortex brain samples, including 31 NeuN+ samples and 31 NeuN- samples. Gasparoni data is available at GEO (GSE66351). We processed the Mendizabal data by extracting a subset of DNAm sites that corresponds to the specific locations matched with scMD and EpiSCORE references respectively. We prepossessed the Gasparoni, and Guintivano data using the minfi package[7]. We evaluated the performance of scMD and EpiSCORE using total MAE comparing estimated and measured fractions.

*Bulk brain DNAm datasets for validation.* The bulk DNA methylation (DNAm) data for the Mount Sinai Brain Bank (MSBB)[25] were obtained from Synapse (ID: syn21347197). This data encompasses 201 tissue samples derived from the parahippocampal gyrus region of the brain (Brodmann area 36), and processed using the Illumina EPIC platform. We first deconvolved the MSBB DNAm data, subsequently examining the relationship between cellular fractions and the Braak stage of Alzheimer's disease (AD).

We also used brain DNAm data from the Religious Orders Study (ROS), specifically from the dorsolateral prefrontal cortex (DLPFC) tissue of 49 senior donors. This dataset incorporates both bulk DNAm data, captured through a 450k array as described by De Jager et al.[22], and measured cell-type

fractions as reported by Patrick et al.[39]. The study measured the proportions of four distinct cell types, namely astrocyte, microglia, neuron, and oligodendrocyte. Note that we excluded endothelial cells since prior studies confirmed their poor quality of measured cell counts[10].

In addition, paired bulk DNAm and RNA data from the nucleus accumbens (NAc) were obtained from public repositories, GSE147040 and GSE171936, respectively. The DNAm data for NAc was profiled using the Infinium MethylationEPIC microarray following the guidelines provided by the manufacturer. The raw idat files were subsequently processed and normalized using the minfi R package.

**Single-cell RNA-sequencing references to deconvolve bulk RNA-seq data**. In our prior research, we compiled a selection of scRNA-seq reference datasets[10]. For the present study, we utilized the brain scRNA-seq data curated by STAB[40] from three studies: Darmanis et al.[41], Hodge et al.[42], and Habib et al.[43]. Deconvolution results were then obtained via EnsDeconv. The single deconvolution methods implemented in EnsDeconv to derive the RNA estimated fraction extend beyond those used in DNAm EnsDeconv. Notably, we excluded deconvolution methods initially intended for DNAm deconvolution, including Houseman et al.[33] and RPC. This included an additional hybrid scale method–dtangle[44]–and two deconvolution approaches specifically designed for scRNA-seq references: MuSiC[5] and Bisque[6]. These methods are not used in the DNAm deconvolution due to their methodological incompatibility with DNAm data. The datasets analyzed in this study were derived from previously published research that received ethical approval.

**Statistics and reproducibility**. In our analysis of scDNAm datasets, we first applied Fisher's exact test to the aggregated, subsetted scDNAm data. Subsequently, we selected the top 100 markers for each cell type as ranked by the Fisher-exact test $p$-value to construct distinct signatures. These signatures were then utilized to estimate the cell type proportions in various bulk datasets. We also compared with the EpiSCORE-derived signature. The resources needed to generate the EpiSCORE reference matrix, including the code and data, were directly sourced from the Code Ocean repository: https://codeocean.com/capsule/2549317/tree/v3. Our investigation extended to comparing the proportion estimates derived from scMD, HiBED, and EpiSCORE. We systematically applied each method across all samples within multiple datasets, including those from Mendizabal ($n = 45$), Guintivano ($n = 58$), Gasparoni ($n = 62$), ROS ($n = 49$), MSBB ($n = 201$), and NAc ($n = 211$). This comprehensive application allowed us to obtain the estimated proportions for each dataset. To validate our findings, we conducted correlation tests. These tests compared the estimated proportions from each method with the actual measured fractions for the Mendizabal, Guintivano, Gasparoni datasets, and the RNA-estimated fractions for NAc. To assess our model's accuracy, we conducted Friedman-Nemenyi posthoc test on MAE to compare scMD, HiBED, and EpiSCORE. We also performed a one-sided Diebold-Mariano test to compare the absolute errors in estimated versus measured fractions for scMD, HiBED, and EpiSCORE. Additionally, we delved into examining the differential cellular fractions associated with varying phenotypes in the MSBB datasets, considering the results from scMD, HiBED, and EpiSCORE. Further enhancing our analysis, we utilized CellDMC to identify cell type-specific differentially methylated cytosines (CTS-DMCs) based on the fractions estimated by scMD.

**Reporting summary**. Further information on research design is available in the Nature Portfolio Reporting Summary linked to this article.

## Data availability

The processed signatures from our research are accessible at the GitHub repository: https://github.com/randel/scMD/tree/main/Processed_data_450k850k. The Lee scDNAm data is obtainable from the NCBI's GEO database under the accession number GSE130711. The Tian scDNAm data can be accessed at GSE215353. The Guintivano DNAm data is available through the Bioconductor package FlowSorted.DLPFC.450k. The Gasparoni DNAm data is publicly accessible via GSE66351, and the Mendizabal DNAm data can be found under GSE108066. The NAc bulk DNAm and RNA data are listed under GSE147040 and GSE171936,. The bulk data for ROS and MSBB, along with their clinical data, are accessible via the AD Knowledge Portal: ROS data (syn3219045) and MSBB data (syn3159438). Source data underlying main figures are provided in Supplementary Data 5–9.

## Code availability

scMD is publicly hosted on GitHub (https://github.com/randel/scMD)[45]. EnsDeconv is available on GitHub (https://github.com/randel/EnsDeconv). EpiSCORE is downloaded from https://github.com/aet21/EpiSCORE. HiBED is downloaded from https://github.com/SalasLab/HiBED.

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

## Acknowledgements

This research was funded in part through NIH's R01AG080590, R03OD034501, R01MH123184, and UL1TR001857. This research was supported in part by the University of Pittsburgh Center for Research Computing through the resources provided. The results published here are in part based on data obtained from the AD Knowledge Portal. Study data were provided by the Rush Alzheimer's Disease Center, Rush University Medical Center, Chicago. Data collection was supported through funding by NIA grants P30AG10161 (ROS), R01AG15819 (ROSMAP; genomics and RNAseq), U01AG46152 (ROSMAP AMP-AD, targeted proteomics), U01AG61356 (whole-genome sequencing, targeted proteomics, ROSMAP AMP-AD) and the Illinois Department of Public Health (ROSMAP). The MSBB data were generated from postmortem brain tissue collected through the Mount Sinai VA Medical Center Brain Bank and were provided by Dr. Eric Schadt from Mount Sinai School of Medicine.

## Author contributions

This study was conceived of and led by J.W. Jointly with J.W. and C.M., M.C. designed the algorithm, implemented the scMD software, and led the data analyses. J.Z. helped provide data and scientific insights on scDNAm. M.C., J.W., C.M., and J.Z. wrote the paper.

## Competing interests

The authors declare no competing interests.
