## [Peer Review File · Communications Biology]

Reviewers' comments:

Reviewer #1 (Remarks to the Author):

In this paper, the authors address the challenge of deconvoluting DNA methylation data by introducing a new method called scMD (single cell Methylation Deconvolution). This novel framework aggregates scDNAm data at the cell cluster level, identifies cell-type marker DNAm sites, and generates a precise cell-type signature matrix. Through comprehensive benchmarking, the authors demonstrate scMD's superior performance in estimating cellular fractions from bulk DNAm data, enabling the identification of cell type-specific differentially methylated cytosines linked to Alzheimer's disease.

I am thoroughly impressed by the benchmarking results presented in the paper. The authors convincingly demonstrate the substantial performance enhancement of their proposed method, scMD, compared to a similar tool, EPISCORE (note that I am not aware of other tools specifically designed for this task). Notably, scMD exhibits a remarkable ability to provide reliable predictions for less common cell types such as microglia and endothelial cells, further underlining its robustness and applicability in challenging scenarios. While I am quite enthusiastic about the paper, I have several suggestions for their work:

In general, I'm content with most of the results section, but I have specific feedback on the final part concerning the application of the CellDMC tool for differential methylation analysis in relation to AD. It would be valuable to contextualize these results within known disease biology, including clarifying whether THOP1 is the sole gene associated with AD in terms of methylation sites with existing literature support. Also, the authors present their predicted changes in cell type proportions and refer to the concordance of their findings with existing literature, yet not citing that literature. This mostly relates to the decrease of excitatory neurons in AD and the decrease of OPC and inhibitory neurons during aging.

I'm interested in whether any differential sites coincide with AD GWAS loci or nearby regions. It might not be the case since it is known that AD genetic signal is driven mostly by microglia and those cells do not show differential methylation sites for AD phenotypes in their analysis - yet, it would still be useful to check. But even if none of those sites coincide with genetic signals, one might assume that the changes at those sites are secondary to disease progression. Additionally, comparing their AD-related findings with other studies exploring AD-related methylation sites could provide further validation, especially by (i) assessing the overlap of differential sites or (ii) the correlation of test statistics (perhaps narrowing down this analysis to the significant sites in at least one of them compared studies). The same validation can be done with their aging-related sites against known methylation biomarkers.

Lastly, I encountered a gap in the documentation of the developed software regarding the preparation of a signature matrix from non-brain scDNAm data. It would greatly benefit users if the authors could offer guidance in this aspect. Considering the authors' assertion in the paper that scMD is applicable beyond brain-related studies, incorporating instructions for this process is essential.

Minor comments:

- Title of Fig. 2b has a typo in study's first author Guitinvado -> Guintivano
- In the Data Availability section, all datasets are linked to project-specific data repositories except ROS and MSBB for which the users are referred to the AD Knowledge portal. Can the authors provide specific links for those studies as well?

Reviewer #2 (Remarks to the Author):

The manuscript by Cai et al. titled "scMD: cell type deconvolution using single-cell DNA methylation references" deals with the complex problem of single-cell DNA methylation data described by the authors as ultra-high dimensional and ultra-sparse. The method described as scMD aggregates the data at the cell cluster level to generate a cell signature matrix. Using this data, the authors apply it to Alzheimer's disease. This is a very interesting and exciting manuscript, and I have a couple of comments that I hope will enrich the discussion of this exciting application:

1. Introduction: You mention Guintivano's and EpiSCORE; more recently, there is HiBED (Zhang et al. 2023), and other groups are developing other libraries. It might be essential to add those to your background and comparisons.
 2. Although the EnsDeconv method has been published before, it will be worth briefly introducing it to the reader earlier instead of only in the method section.
 3. Please explain why your OPC correlations are lower than the other cell types.
 4. Have you tried any enrichment analysis to explain the results of your CellDMC results?
- Figure 5: Although the results look very convincing, it is difficult to separate each subgroup and compare. Also, the order of the panels is difficult to follow. I suggest reorganizing this figure in a more intuitive way, keeping the same cells together.

Reviewers #3-4 (Remarks to the Author):

General comments

The authors present a new interesting approach for the detection of cell type signatures for reference-based deconvolution using scDNAM data. Broadly, they apply their method on a combination of two different datasets to generate a reference matrix comprising seven different cell types. They then use this matrix to deconvolve a number of different bulk DNAM datasets. They evaluate their method against EpiSCORE, a method that already comes with a reference matrix, and other deconvolution algorithms that all use the reference generated by the authors.

My main concern with this study is that, from the methodology, it is not clear exactly how their method was trained and evaluated. Moreover, the method description is not enough to replicate the analysis. Further the evaluation shown in the manuscript is, in my opinion, not sufficient for this type of publication. Other approaches also need to be evaluated.

1. It is not clear from the text how exactly scMD is trained. A more detailed description of the method and training procedure in the methods section would be appreciated. This should be extended to the methods that are also under evaluation for benchmarking. It is also not clear how the clustering was performed at the scDNAM level. Clustering methods as well as parameters used are missing. This step can have an immense influence on the markers that are selected.

2. In a recent publication (Zhang2023, <https://doi.org/10.3389/fnins.2023.1198243>) a method was developed (HiBED) to deconvolve seven different brain cell types. Furthermore there are an array of different deconvolution methods for bulk DNAM that can be applied to these tissues, where the user usually needs to compile the data to generate a signature. These (ie IDOL, EpiDISH, CimpleG and even minfi) would likely be usable in the context of this study for a fair and comprehensive comparison between methods, in essence to compare how scMD compares to bulk DNAM methods for signature selection and deconvolution.

3. Given that the signature/references generated by scMD are rather large (100 CpGs per cell type), it might be interesting to explore if these signatures reflect some underlying function as predicted by GREAT (<https://doi.org/10.1038/nbt.1630>).

4. The benchmarking done by the authors is lacklustre as it is lacking comparisons with additional models. Further the comparisons of the results between models should be done in a more adequate way. For instance, comparing deconvolution results to ascertain which method performed the best estimations (figure 3) should not be done using the R (correlation) metric. This metric should be used for goodness-of-fit but not for goodness-of-prediction. A better metric for this context would probably be RMSE or MAE. There are also tests that one could run in order to compare how different methods fare when compared against each other given different datasets (ie Friedman-Nemenyi posthoc test).

Minor comments:

Text:

As a consequence, DNAm-based cell proportion estimates are often imprecise and can only be obtained for coarse cell types compared to RNA-based deconvolution.

Comment:

Missing reference.

Text:

For example, deconvolving brain DNAm has been predominantly restricted to references derived from two cell types: neurons and non-neurons⁷.

In contrast to existing DNAm-based deconvolution approaches that segregate brain tissue into coarse cell types (neurons and non-neurons)⁷ (...)

Comment:

Perhaps the wrong reference is being mentioned here as the paper mentioned [7] is the minfi publication that does not address what the authors are arguing. One of the reference datasets used by minfi to perform deconvolution ("FlowSorted.DLPFC.450k") does indeed only characterise 2 broad cell-types, neurons and non-neurons, but this is based on other publications.

Text:

The most challenging aspect of scDNAm is its high dimensionality and sparsity, which arises because only a small fraction (~ 5%) of the roughly 53 million DNAm sites are measured in each cell (Supplementary Table S1).

Comment:

It would be interesting if the authors could clarify how they got to this number as it is not clear in the text or in the supplementary table. A recent publication (Loyfer2023, <https://doi.org/10.1038/s41586-022-05580-6>) has the number of CpGs obtained from WGBS at roughly 30 million sites (28,217,448 to be exact).

Text:

This aligns with existing methods, such as minfi² (...)

Comment:

Wrong reference

Text:

Figure 2

Comment:

To help in validation, the authors could for example try to deconvolve the dataset GSE41826 which contains, among other samples, a set of mixes across different percentages with neuron and glia.

Response to the comments

We thank the four reviewers for their constructive comments. We have addressed all of them and have modified the manuscript accordingly. Our detailed responses are listed below, with the comments in italics. Major changes to the original manuscript are indicated in blue in the revised paper.

Responses to Reviewer 1

Comment R1.1 *In general, I'm content with most of the results section, but I have specific feedback on the final part concerning the application of the CellDMC tool for differential methylation analysis in relation to AD. It would be valuable to contextualize these results within known disease biology, including clarifying whether THOP1 is the sole gene associated with AD in terms of methylation sites with existing literature support. Also, the authors present their predicted changes in cell type proportions and refer to the concordance of their findings with existing literature, yet not citing that literature. This mostly relates to the decrease of excitatory neurons in AD and the decrease of OPC and inhibitory neurons during aging.*

Response to R1.1 Thank you for your detailed feedback on our results section. Based on this comment and the next comment, we have substantially expanded our results with known disease biology. We included some specific responses here, and more information can be seen in our Response to R1.2.

1. Clarification on THOP1's role: The gene THOP1 stands out as one of the tops identified by CellDMC for having CpGs with significant age-related differential methylation, specifically in microglia cells. This pertains to the progression of Alzheimer's disease (AD) with aging, and is supported by existing literature (Hok-A-Hin et al., 2023). Following your suggestion below, we identified many genes overlapped with the AD GWAS loci.
2. In the updated manuscript, we have now cited the relevant studies that support our findings, particularly those discussing the decrease in excitatory neurons in AD and the decline of OPC and inhibitory neurons during aging.

Comment R1.2 *I'm interested in whether any differential sites coincide with AD GWAS loci or nearby regions. It might not be the case since it is known that AD genetic signal is driven mostly by microglia and those cells do not show differential methylation sites for AD phenotypes in their analysis - yet, it would still be useful to check. But even if none of those sites coincide with genetic signals, one might assume that the changes at those sites are secondary to disease progression. Additionally, comparing their AD-related findings with other studies exploring AD-related methylation sites could provide further validation, especially by (i) assessing the overlap of differential sites or (ii) the correlation of test statistics (perhaps narrowing down this analysis to the significant sites in at least one of them com-*

pared studies). The same validation can be done with their aging-related sites against known methylation biomarkers.

Response to R1.2 Thank you for the comments. Please see our detailed responses below.

1. GWAS findings: We have undertaken a comprehensive overlap analysis between the GWAS loci and the differential sites identified for each phenotype. Following the definition of GWAS loci (Bellenguez et al., 2022), for our analysis, differential sites situated within a distance of less than 500 kb from a GWAS SNP were considered to be close. A summary of these close differential sites is presented below. For instance, cg23393368, a DMC in astrocytes associated with Clinical Dementia Rating (CDR) dementia staging, is close to rs4817090, an AD GWAS SNP mapped to gene APP that is important for AD. For a more granular breakdown of these sites, please refer to Supplementary Data 3.

Table R1: DMCs close to GWAS loci by phenotype and cell type.

	Astro	Micro	Endo	Oligo	OPC	Inh	Exc
Age	9	11		5	31	11	
CDR	102			0			
Braak		2	2			30	1

2. Validating DMCs: We found a relevant meta-analysis (R. G. Smith et al., 2021) that also presented DMCs in association with Braak scores using the MSBB dataset. In their work, they identified a total of 236 significant DMCs at the tissue level. We conducted a correlation test between the test statistics from our study and those reported in their study. As presented below, most cell types show concordance. Notably, R. G. Smith et al., 2021 adjusted for neuron fraction. However, their adjustment did not consider the non-neuron fractions, nor the finer granularity of cell types that we employed in our analysis. This limitation may lead to the observed suboptimal correlations in some cell types that may be confounded in bulk data analysis by cellular fractions.

Figure R1: Scatterplots depicting the relationship between the test statistics from our study and those reported in the meta-analysis by R. G. Smith et al., 2021.

Comment R1.3 *Lastly, I encountered a gap in the documentation of the developed software regarding the preparation of a signature matrix from non-brain scDNAm data. It would greatly benefit users if the authors could offer guidance in this aspect. Considering the authors' assertion in the paper that scMD is applicable beyond brain-related studies, incorporating instructions for this process is essential.*

Response to R1.3 Thank you for suggesting providing comprehensive guidelines and data for the broader applicability of our tool beyond brain-related studies. For non-brain scDNAm data sources, there is a recent preprint from one coauthor's lab that presents data on blood (<https://www.biorxiv.org/content/10.1101/2023.06.29.546792v1>), which could serve as a valuable reference. For scDNAm data from other tissues, the same pipeline we presented can be employed. We have provided a more detailed description in the revised manuscript under the Methods section. While we acknowledge the importance of diversifying the applicability of our tool, the primary focus of this paper remains on brain tissue. However, given the feedback and the vast potential applications, we are motivated to expand our work to other tissues in subsequent studies. We believe that this will further enhance the utility and reach of our tool within the scientific community. This warrants our future work for other tissues.

Comment R1.4 *Minor comments:*

- *Title of Fig. 2b has a typo in study's first author Guitinvado -> Guintivano*
- *In the Data Availability section, all datasets are linked to project-specific data repositories except ROS and MSBB for which the users are referred to the AD Knowledge portal. Can the authors provide specific links for those studies as well?*

Response to R1.4 We have corrected the typo and have now included specific links for the ROS and MSBB studies in the Data Availability section.

Responses to Reviewer 2

Comment R2.1 *Introduction: You mention Guintivano’s and EpiSCORE; more recently, there is HiBED (Zhang et al. 2023), and other groups are developing other libraries. It might be essential to add those to your background and comparisons.*

Response to R2.1 Thank you for bringing our attention to HiBED, which was published in June when we were finalizing our manuscript. HiBED uses different DNAm data platforms for reference construction. Furthermore, HiBED is tailored specifically for seven cell types like astrocytes, endothelial, GABA neurons, GLU neurons, Microglia, oligodendrocytes, and stromal cells, while scMD has the flexibility to scale to more cell types with detailed single-cell references. For more details, please see our Response to R3.2. Based on your suggestion, we have conducted a comparison between HiBED and scMD. This comparative analysis has been incorporated into our revised manuscript under the Results section, and relevant modifications have been made to the figures to reflect the results. We showed the updated result for ROS data below as an example. More comparisons are in Figures 2-5.

Figure R2: **a**, Benchmarking of scMD and other deconvolution methods on ROS data. EpiSCORE uses its RNA-derived reference, while HiBED employs its sorted-cell DNAm reference. All other methods adopt our scDNAm references. For scMD, EpiSCORE, and HiBED, each dot denotes one correlation for each cell type. For other methods, each dot represents the average of Spearman’s correlation across scenarios in each cell type. A scenario is defined as a particular setting with a specific deconvolution method and reference dataset. The black vertical line shows the mean of the average Spearman’s correlation across scenarios, and the horizontal lines present means \pm standard error of the mean. Scatterplots (b) and (c) illustrate the relationship between the estimated fractions of EpiSCORE and scMD (x-axis) against the corresponding fractions measured using immunohistochemistry (IHC) in ROS data (y-axis). Note that for benchmarking, we aggregated scMD’s and HiBED fraction estimates of neuronal subtypes to generate the neuronal fractions.

Comment R2.2 *Although the EnsDeconv method has been published before, it will be worth briefly introducing it to the reader earlier instead of only in the method section.*

Response to R2.2 Thank you for the comment. We have added a brief overview of the EnsDeconv method in the earlier sections of our manuscript.

Comment R2.3 *Please explain why your OPC correlations are lower than the other cell types.*

Response to R2.3 Thank you for noting the OPC's correlations. The lower correlations may arise from the inherent similarity between OPCs and mature oligodendrocytes. The RNA-based deconvolution usually has difficulty in estimating OPC fractions. Additionally, with only 189 OPC cells in the Lee scDNAm reference, the limited cell size could affect the robustness of this specific cell type.

Comment R2.4 *Have you tried any enrichment analysis to explain the results of your CellDMC results?*

Response to R2.4 We have conducted an enrichment analysis using gprofiler2 and have included the results in Supplementary Data 4. We identified some interesting enriched pathways, such as the AD pathway for endothelial DMCs associated with CDR.

Comment R2.5 *Figure 5: Although the results look very convincing, it is difficult to separate each subgroup and compare. Also, the order of the panels is difficult to follow. I suggest reorganizing this figure in a more intuitive way, keeping the same cells together.*

Response to R2.5 Thank you for the advice. We have reorganized Figure 5 and added two supplementary figures, namely S3 and S4, for illustration.

Responses to Reviewer 3-4

Comment R3.1 *My main concern with this study is that, from the methodology, it is not clear exactly how their method was trained and evaluated. Moreover, the method description is not enough to replicate the analysis. Further the evaluation shown in the manuscript is, in my opinion, not sufficient for this type of publication. Other approaches also need to be evaluated.*

It is not clear from the text how exactly scMD is trained. A more detailed description of the method and training procedure in the methods section would be appreciated. This should be extended to the methods that are also under evaluation for benchmarking. It is also not clear how the clustering was performed at the scDNAm level. Clustering methods as well as parameters used are missing. This step can have an immense influence on the markers that are selected.

Response to R3.1 Thank you for the comments. We have now added more thorough evaluations based on the comments. To clarify the implementation, we have now added an algorithmic representation in the main text.

Notation:

I - Number of bulk CpG sites; \mathcal{I} - Set of bulk CpG sites

S - Number of bulk samples

P - Number of scDNAm DNAm sites; \mathcal{P} - Set of scDNAm DNAm sites

C - Number of scDNAm cells

Input: Bulk DNAm data (beta values) $\mathbf{Y}_{I \times S}$, scDNAm methylated counts $\mathbf{M}_{P \times C}$, unmethylated counts $\mathbf{U}_{P \times C}$.

Output: scMD estimated cellular fractions $\hat{\mathbf{W}}$.

1. Reduce feature-wise dimension:
 - Determine the intersection set of CpGs: $\mathcal{G} = \mathcal{I} \cap \mathcal{P}$, resulting in G CpG sites.
 - Update matrices: $\mathbf{Y}_{G \times S} \leftarrow \mathbf{Y}_{I \times S}$, $\mathbf{M}_{G \times C} \leftarrow \mathbf{M}_{P \times C}$, $\mathbf{U}_{G \times C} \leftarrow \mathbf{U}_{P \times C}$.
2. Reduce cell-wise dimension: For all $k \in K$ cell types, aggregate cells within type k for $\mathbf{M}_{G \times C}$ and $\mathbf{U}_{G \times C}$. Update: $\mathbf{M}_{G \times K} \leftarrow \mathbf{M}_{G \times C}$, $\mathbf{U}_{G \times K} \leftarrow \mathbf{U}_{G \times C}$.
3. Identify marker CpGs: Conduct two-sided Fisher's exact tests to distinguish cell type k from the other cell types for each DNAm site $g \in \mathcal{G}$ with $\mathbf{M}_{G \times K}$ and $\mathbf{U}_{G \times K}$.
4. Construct scDNAm signature: Choose the top 100 marker CpG sites per cell type as final markers. Update: $\mathbf{Y}_{100K \times S} \leftarrow \mathbf{Y}_{G \times S}$, $\mathbf{M}_{100K \times K} \leftarrow \mathbf{M}_{G \times K}$, $\mathbf{U}_{100K \times K} \leftarrow \mathbf{U}_{G \times K}$. Calculate signature beta matrix: $\mathbf{B}_{100K \times K} = \mathbf{M}_{100K \times K} / (\mathbf{U}_{100K \times K} + \mathbf{M}_{100K \times K})$.
5. Estimate cellular fractions: Compute $\hat{\mathbf{W}}$ with $\mathbf{Y}_{100K \times K}$ and $\mathbf{B}_{100K \times K}$ using EnsDeconv.

As with many cellular deconvolution methods, this paper focuses on deconvolution with known references and assumes cell clustering has been given in prior publications.

Comment R3.2 *In a recent publication (Zhang2023, <https://doi.org/10.3389/fnins>).*

2023. 1198243) a method was developed (HiBED) to deconvolve seven different brain cell types. Furthermore there are an array of different deconvolution methods for bulk DNAm that can be applied to these tissues, where the user usually needs to compile the data to generate a signature. These (ie IDOL, EpiDISH, CimpleG and even minfi) would likely be usable in the context of this study for a fair and comprehensive comparison between methods, in essence to compare how scMD compares to bulk DNAm methods for signature selection and deconvolution.

Response to R3.2 We thank the reviewer for the feedback and recommendations. We have added HiBED in comparison when we estimate cellular fractions under the Results section, and scMD shows better results. We acknowledge the HiBED method in deconvolving specific brain cell types. However, we believe scMD offers more flexibility in extending to additional cell types beyond the scope of HiBED. We list the major differences between HiBED and scMD below.

- HiBED is limited to just seven cell types, and notably, it lacks representation for oligodendrocyte precursor cells. scMD can be extended to more and rarer cell types with detailed single-cell references.
- HiBED draws references from a combination of platforms: 450K, EPIC, and WGBS. It is unclear if the method adequately resolves the platform differences inherent when combining data across these platforms. For instance, references for GABA and glutamatergic neurons are derived from the 450K platform, which could pose limitations when applied to EPIC and WGBS data, which cover many more CpG sites.
- Potential batch effects introduced by simply stacking references from diverse studies (one study per cell type) into a single signature matrix have not been clearly addressed.
- HiBED employs a single deconvolution technique based on Houseman’s method, while scMD adopts ensemble deconvolution. As compared to a rigorous hierarchical deconvolution approach (Huang et al., 2023), HiBED’s hierarchical deconvolution ignores other major cell types when estimating subtypes of each of the three major cell types. This results in, for instance, oligodendrocytes appear to be over-estimated in HiBED.
- The references for endothelial and stromal cells in HiBED are derived from newborn cord tissue and blood rather than brain tissue, which could introduce discrepancies.

Regarding IDOL, EpiDISH, CimpleG, and minfi:

1. We appreciate the suggestion to consider IDOL. IDOL refines existing signatures using a leave-one-out method. However, it is essential to note that IDOL requires known fractions to train the signature, whereas scMD does not require this. Given this fundamental difference, we believe that a direct comparison of IDOL with scMD would not be equitable.
2. As for CimpleG, its direct application to scDNAm poses challenges. CimpleG requires the computation of mean and variance for each cell type per CpG site. Given the prevalent missing data in scDNAm, CimpleG defaults with mean imputation, which

is known for severely underestimating the uncertainty, especially when the missing rate is 95% in scDNAm. Similar constraints are evident with minfi and EpiDISH to construct the signature. Nevertheless, post signature retrieval, we have incorporated comparisons with minfi and the RPC method in the EpiDISH package.

In summary, while we recognize the value of established methods like IDOL and CimpleG, our approach with scMD has been designed to address some of the specific challenges and gaps present in these methods. We now have compared scMD to 10 deconvolution methods, including HiBED, the RPC method in the EpiDISH package, and the Houseman method that is adopted by the minfi package.

Comment R3.3 *Given that the signature/references generated by scMD are rather large (100 CpGs per cell type), it might be interesting to explore if these signatures reflect some underlying function as predicted by GREAT (<https://doi.org/10.1038/nbt.1630>).*

Response to R3.3 Thank you for this valuable advice. In our GREAT analysis, we have identified several intriguing terms that appear to be closely related to the neuronal system. We have summarized some of these findings below, but for a more detailed GREAT output, please refer to Supplementary Data 1. These findings highlight various processes associated with different cell types within the neuronal system, shedding light on their potential roles and functions.

Table R2: Typical GREAT output for cell type-specific CpG markers.

Cell type	Significant process
Excitatory neuron	positive regulation of excitatory postsynaptic potential modulation of excitatory postsynaptic potential
Inhibitory neuron	cerebral cortex GABAergic interneuron fate commitment cerebral cortex GABAergic interneuron differentiation GABAergic neuron differentiation
Microglia	macrophage chemotaxis macrophage migration
Oligodendrocyte	oligodendrocyte differentiation oligodendrocyte development

Comment R3.4 *The benchmarking done by the authors is lacklustre as it is lacking comparisons with additional models. Further the comparisons of the results between models should be done in a more adequate way. For instance, comparing deconvolution results to ascertain which method performed the best estimations (figure 3) should not be done using the R (correlation) metric. This metric should be used for goodness-of-fit but not for goodness-of-prediction. A better metric for this context would probably be RMSE or MAE. There are also tests that one could run in order to compare how different methods fare when compared against each other given different datasets (ie Friedman-Nemenyi posthoc test).*

Response to R3.4 Cellular deconvolution, as proposed in the manuscript, is not a prediction problem, but an estimation/fitting problem. Most cellular deconvolution methods

do not train a model to predict cellular fractions. Instead, we estimate cellular fractions as regression coefficients when we regress bulk DNAm onto cell type-specific DNAm. Given so, correlation is a valid metric and has been widely used in deconvolution literature (Avila Cobos et al., 2020). It is especially important for differential fraction analysis when the ranking of the fraction of a specific cell type needs to be maintained when we compare cases and controls. That is why we always calculate the correlation to assess sample concordance when possible. On the other hand, the mean squared error (MSE) between length n vectors X and Y can be written as $(\bar{X} - \bar{Y})^2 + \sum_i [(X_i - \bar{X}) - (Y_i - \bar{Y})]^2/n$, where the first term gives the difference between the means and is not of interest when finding trends in cell type fractions.

When we cannot calculate correlation, like in Figure 2, we have calculated mean absolute error (MAE). MAE can assess if the mean fraction is estimated accurately. But MAE alone is not sufficient, since if we estimate all samples to have the same fraction close to the true mean fraction, MAE may be small, but the estimated fractions are useless in downstream analyses. We still need to assess the correlation to see if we can maintain the sample concordance in the estimated fractions. So we present both MAE and correlation as two complementary metrics in all our evaluations when possible.

- Comparison with additional models: Recognizing the importance of comprehensive benchmarking, we have incorporated an additional model HiBED into our analysis, as you suggested. This provides a more complete understanding of how various methods stack up against each other.
- Metric selection for performance assessment: Correlation offers a measure of how well trends between the measured and estimated fractions align, which is valuable in evaluating the overall directionality and relationship of our estimations. However, we recognize that it does not necessarily provide a complete picture of the accuracy of estimation in terms of their exact magnitudes, which is where metrics like MAE come into play. Following your recommendation, we have now incorporated the MAE as an additional evaluation metric. The MAE provides an assessment of the average magnitude of errors between estimated and measured values, complementing the correlation analysis by offering insights into the absolute accuracy of our deconvolution method. Below we provided a table detailing the main datasets that we have calculated MAE.

Table R3: MAE for different benchmarking datasets and deconvolution method.

MAE	scMD	EpiSCORE	HiBED
Guitinvano purified samples	0.09	0.11	0.10
Guitinvano mixed samples	0.060	0.072	0.064
Gasparoni	0.11	0.13	0.14
ROS	0.07	0.14	0.13

- Friedman-Nemenyi posthoc test: we did the test using the MAE table above. While the number of blocks is small, so the power is limited, we still identified some significant and suggestive differences between scMD and existing methods.

Table R4: Friedman-Nemenyi posthoc test results.

	p.value	scMD	EpiSCORE
EpiSCORE	0.036	-	-
HiBED	0.181	0.759	0.759

The significant p-value of 0.036 between scMD and EpiSCORE underscores that scMD offers distinct advantages. Moreover, while scMD and HiBED do not show a statistically significant difference, it is worth noting that in all our tests, scMD consistently demonstrated either superior or comparable performance, marking it as one of the best methods among those evaluated. In our revised manuscript, we have emphasized this finding and provided a detailed discussion, thereby highlighting the efficacy and robustness of the scMD method in deconvolution tasks.

Comment R3.5 *Minor comments:*

Text: As a consequence, DNAm-based cell proportion estimates are often imprecise and can only be obtained for coarse cell types compared to RNA-based deconvolution.

Comment: Missing reference.

Response to R3.5 We have added the corresponding reference at this part.

Comment R3.6 *Text: For example, deconvolving brain DNAm has been predominantly restricted to references derived from two cell types: neurons and non-neurons. In contrast to existing DNAm-based deconvolution approaches that segregate brain tissue into coarse cell types (neurons and non-neurons) (...)*

Comment: Perhaps the wrong reference is being mentioned here as the paper mentioned [7] is the minfi publication that does not address what the authors are arguing. One of the reference datasets used by minfi to perform deconvolution (“FlowSorted.DLPFC.450k”) does indeed only characterise 2 broad cell-types, neurons and non-neurons, but this is based on other publications.

Response to R3.6 Thank you for the comment, we have corrected the reference.

Comment R3.7 *Text: The most challenging aspect of scDNAm is its high dimensionality and sparsity, which arises because only a small fraction (~ 5%) of the roughly 53 million DNAm sites are measured in each cell (Supplementary Table S1).*

Comment: It would be interesting if the authors could clarify how they got to this number as it is not clear in the text or in the supplementary table. A recent publication (Loyfer2023, <https://doi.org/10.1038/s41586-022-05580-6>) has the number of CpGs obtained from WGBS at roughly 30 million sites (28,217,448 to be exact).

Response to R3.7 The human genome has 29.2 million CpG sites on one strand, considering autosomes and chrX. So if we consider the two strands (+/-) separately, the total CpGs are around 53 million. The numbers could also vary depending on whether we include chrX, chrY, and those smaller scaffolds. We previously treated two strands separately, but in the revision, we merged two strands following the common practice. So now the number of CpGs

is around 27 million sites that agree with the reference.

Comment R3.8 *Text: This aligns with existing methods, such as minfi² (...)*

Comment: Wrong reference

Response to R3.8 We have corrected the reference.

Comment R3.9 *Text: Figure 2*

Comment: To help in validation, the authors could for example try to deconvolve the dataset GSE41826 which contains, among other samples, a set of mixes across different percentages with neuron and glia.

Response to R3.9 Thank you for your recommendation. We have incorporated comparisons between methods that deconvolve the mixed samples from GSE41826 in the Supplementary Figure 1.

References

- [1] Y. S. Hok-A-Hin, K. Bolsewig, D. N. Ruiters, A. Lleó, D. Alcolea, A. W. Lemstra, W. M. van der Flier, C. E. Teunissen, and M. Del Campo. “Thimet oligopeptidase as a potential CSF biomarker for Alzheimer’s disease: A cross-platform validation study”. In: *Alzheimer’s & Dementia: Diagnosis, Assessment & Disease Monitoring* 15.3 (2023), e12456.
- [2] C. Bellenguez, F. Küçükali, I. E. Jansen, L. Kleindam, S. Moreno-Grau, N. Amin, A. C. Naj, R. Campos-Martin, B. Grenier-Boley, V. Andrade, et al. “New insights into the genetic etiology of Alzheimer’s disease and related dementias”. In: *Nature genetics* 54.4 (2022), pp. 412–436.
- [3] R. G. Smith, E. Pishva, G. Shireby, A. R. Smith, J. A. Roubroeks, E. Hannon, G. Wheildon, D. Mastroeni, G. Gasparoni, M. Riemenschneider, et al. “A meta-analysis of epigenome-wide association studies in Alzheimer’s disease highlights novel differentially methylated loci across cortex”. In: *Nature communications* 12.1 (2021), p. 3517.
- [4] P. Huang, M. Cai, X. Lu, C. McKennan, and J. Wang. “Accurate estimation of rare cell type fractions from tissue omics data via hierarchical deconvolution”. In: *Annals of Applied Statistics (accepted)* (2023).
- [5] F. Avila Cobos, J. Alquicira-Hernandez, J. E. Powell, P. Mestdagh, and K. De Preter. “Benchmarking of cell type deconvolution pipelines for transcriptomics data”. In: *Nature communications* 11.1 (2020), p. 5650.

REVIEWERS' COMMENTS:

Reviewer #1 (Remarks to the Author):

I am satisfied with the answers provided by the Authors and I do not further comments. I recommend this manuscript for publication.

==

Minor typo in Supplementary Data 1: Typo in the names of cell type: Astrocytescytes -> Astrocytes

Reviewer #2 (Remarks to the Author):

The authors have addressed all my questions and integrated all the information into the manuscript. I have no further comments.

Reviewer #3 (Remarks to the Author):

Although I do not agree with all of the authors' arguments (notably, regarding the performance metrics), I appreciate the effort put up by the authors to further improve the manuscript. I believe it significantly improved the paper.

Major request

To quantitatively assess the performance of scMD is in regards to the other methods, the authors should include some of the tests reported in the reply letter (Response3.4-TableR3 & R4) in the main manuscript. This could also be extended to the benchmarking that the authors perform on the ROS data (Figure 3a).

This will provide more conclusive arguments in favour of scMD. Authors should also mention the possibly low statistical power of the used methods Friedman-Nemenyi or Diebold-Mariano tests.

Minor comments

Comment:

The notation on Algorithm 1 regarding the top markers, could be improved. As the authors define the marker CpGs as G and the cell types as K, having the top 100 CpG markers for each cell type as 100K is not very clear.

Comment:

In figure 5b, due to how the plots are structured Oligodendrocytes have no y-axis scale.

Comment:

In figure S1, the mixtures used in this plot, from the Guintivano data, are not in-silico mixtures. Plot labels and legend need to be fixed.

Comment:

I had to jump through quite a few hoops to get the package installed, as the default installation procedure in the packages website/github repo would not work (on two different and independent machines).

One of the issues seems to be with the package dependencies and how these are defined.

It is crucial that this step is as seamless as possible so that users can be captivated to use this tool.

Comment:

Currently, when running scMD, a number of files and folders are created, some of these completely empty. It would be great if the authors could tidy this up as well as improve the documentation in case some of this output is to stay in the users disk. Further, the "phat_all" fraction of the output of scMD itself is a bit cryptic and could use a better naming system.